

**Acylperoxy radicals during ozonolysis of α-pinene: composition, formation mechanism,**
**and contribution to the production of highly oxygenated organic molecules**
Han Zang[1], Dandan Huang[2], Jiali Zhong[3], Ziyue Li[1], Chenxi Li[1], Huayun Xiao[1], Yue Zhao[1,*]
[1]School of Environmental Science and Engineering, Shanghai Jiao Tong University, Shanghai,
200240, China
[2]Shanghai Academy of Environmental Sciences, Shanghai 200233, China
[3]Division of Environment and Sustainability, Hong Kong University of Science and Technology,
Hong Kong SAR, 999077, China
*Correspondence: Yue Zhao (yuezhao20@sjtu.edu.cn)





**Abstract**
Acylperoxy radicals ($RO_2$) are key intermediates in atmospheric oxidation of organic compounds
and different from the general alkyl $RO_2$ radicals in reactivity. However, direct probing of the
molecular identities and chemistry of acyl $RO_2$ remains quite limited. Here, we report a combined
experimental and kinetic modelling study of the composition and formation mechanisms of acyl
$RO_2$, as well as their contributions to the formation of highly oxygenated organic molecules (HOMs)
during ozonolysis of α-pinene. We find that acyl $RO_2$ radicals account for 67%, 94%, and 32% of
the highly oxygenated $C_7$, $C_8$, and $C_9$ $RO_2$, respectively, but only a few percent of $C_{10}$ $RO_2$. The
formation pathway of acyl $RO_2$ species depends on their oxygenation level. The highly oxygenated
acyl $RO_2$ (oxygen atom number ≥ 6) are mainly formed by the intramolecular aldehydic H-shift (i.e.,
autoxidation) of $RO_2$, while the less oxygenated acyl $RO_2$ (oxygen atom number < 6) are basically
derived from the C-C bond cleavage of alkoxy (RO) radicals containing an α-ketone group or the
intramolecular H-shift of RO containing an aldehyde group. The acyl $RO_2$-involved reactions
explain 50-90% of $C_7$ and $C_8$ closed-shell HOMs and 14% of $C_{10}$ HOMs, respectively. For $C_9$ HOMs,
this contribution can be up to 30%-60%. In addition, acyl $RO_2$ contribute to 50%-95% of $C_{14}$-$C_{18}$
HOM dimer formation. Because of the generally fast reaction kinetics of acyl $RO_2$, the acyl $RO_2$ +
alkyl $RO_2$ reactions seem to outcompete the alkyl $RO_2$ + alkyl $RO_2$ pathways, thereby affecting the
fate of alkyl $RO_2$ and HOM formation. Our study sheds lights on the detailed formation pathways
of the monoterpene-derived acyl $RO_2$ and their contributions to HOM formation, which will help to
understand the oxidation chemistry of monoterpenes and sources of low-volatility organic
compounds capable of driving particle formation and growth in the atmosphere.



## 1. Introduction


Monoterpenes ($C_{10}H_{16}$) comprise an important fraction of nonmethane hydrocarbons in the global
atmosphere (Guenther et al., 2012; Sindelarova et al., 2014) and make a significant contribution to
the secondary organic aerosol (SOA) budget (Pye et al., 2010; Iyer et al., 2021). The presence of
double bond and large molecular size of monoterpenes favor their oxidation reactivity towards $O_3$,
hydroxyl (OH), and nitrate ($NO_3$) radicals (Atkinson et al., 1990; Roger et al., 2004; Kurten et al.,
2015; Kristensen et al., 2016; Bianchi et al., 2019; Berndt, 2022), as well as the formation of low-
volatility products and SOA (Fry et al., 2009; Fry et al., 2014; Zhang et al., 2018; Bianchi et al.,
2019; Molteni et al., 2019; Shen et al., 2022). The organic peroxy radicals ($RO_2$) in the gas-phase
oxidation of monoterpenes can undergo autoxidation and form a class of highly oxygenated organic
compounds (HOM) (Jokinen et al., 2014; Mentel et al., 2015; Berndt et al., 2016; Zhao et al., 2018;
Bianchi et al., 2019; Bell et al., 2021; Berndt, 2022), which are primarily low- or extremely low-
volatility organic compounds (LVOCs and ELVOCs) (Ehn et al., 2014; Bianchi et al., 2019) and
thus play a crucial role in SOA formation and growth.
Significant advances have been made in recent years concerning the monoterpene $RO_2$ autoxidation
and its contribution to HOM formation (Ehn et al., 2014; Berndt et al., 2016; Zhao et al., 2018; Xu
et al., 2019; Lin et al., 2021; Berndt, 2022; Shen et al., 2022). It is recognized that a part of
monoterpene $RO_2$ radicals derived from the traditional ozonolysis channel (i.e., isomerization of
Criegee intermediates, CI) and OH addition channel can autoxidize at a rate larger than 1 $s^{-1}$ and
could be an important contributor to HOM formation (Zhao et al., 2018; Xu et al., 2019; Berndt,
2021). Recently, new reaction channels leading to the $RO_2$ radicals that can undergo fast
autoxidation have been proposed. A quantum chemical calculation study indicated that an excited
CI arising from α-pinene ozonolysis could undergo ring-breaking reactions and directly lead to a
ring-opened $RO_2$ due to the excess energy, which can autoxidize at a rate of ~1 $s^{-1}$ and rapidly form
highly oxidized $RO_2$ with up to 8 oxygen atoms (Iyer et al., 2021). In addition, the minor hydrogen
abstraction channel by OH radicals has been proposed as a predominant pathway to HOM formation
from OH oxidation of α-pinene under atmospheric conditions (Shen et al., 2022).
$RO_2$ species can be simply divided into alkyl $RO_2$ and acyl $RO_2$ (RC(O)OO) according to whether
R is an acyl radical. There are significant differences in the reactivity of these two kinds of $RO_2$.
Firstly, the rate constant of acyl $RO_2$ with NO is in general slightly higher than that of alkyl $RO_2$
(Atkinson et al., 2007; Calvert et al., 2008; Orlando and Tyndall, 2012). For example, the reaction
rate constants of acyl $RO_2$, $CH_3C(O)O_2$, and alkyl $RO_2$, $CH_3CH_2O_2$, with NO have been reported to
be $20 \times 10^{-12}$ $cm^3$ molecule$^{-1}$ $s^{-1}$ and $9.2 \times 10^{-12}$ $cm^3$ molecule$^{-1}$ $s^{-1}$, respectively (Atkinson et al., 2007;



Calvert et al., 2008; Orlando and Tyndall, 2012). Besides, acyl $RO_2$ can react rapidly with $NO_2$ and
form thermally unstable peroxyacyl nitrates ($RC(O)OONO_2$), which have a lifetime of tens of
minutes at room temperature and of days and even months in winter or in the upper atmosphere with
lower temperatures (Atkinson et al., 2007; Orlando and Tyndall, 2012). Although alkyl $RO_2$ radicals
can also react with $NO_2$ and form the alkyl peroxynitrates ($ROONO_2$), they are extremely unstable
and will decompose into $RO_2$ radicals and $NO_2$ in less than 1s (Kirchner et al., 1997; Orlando and
Tyndall, 2012). Lastly, the rate constant of cross-reaction of acyl $RO_2$ ($1.5 \pm 0.3 \times 10^{-11}$ $cm^3$
$molecule^{-1}$ $s^{-1}$) is significantly higher than that of alkyl $RO_2$ ($2 \times 10^{-17}$ - $1 \times 10^{-11}$ $cm^3$ $molecule^{-1}$ $s^{-1}$)
(Villenave and Lesclaux, 1996; Tyndall et al., 2001; Atkinson et al., 2007; Zhao et al., 2018). As a
result, these two kinds of $RO_2$ may play different roles in the autoxidation as well as HOM and
dimer formation.
The quantum calculations revealed that different functional groups in $RO_2$ would lead to
significantly different intramolecular H-shift rates (Otkjær et al., 2018). The C=O and C=C
substituents lead to resonance stabilized carbon radicals and could enhance the H-shift rate constants
by more than a factor of 400. The fast aldehydic H-shift rate contributes to a series of acyl radicals
(RC(O)) with the radical site at the terminal carbonyl carbon, which further produce the acyl $RO_2$
with $O_2$ addition. Many $RO_2$ formed in the oxidation of monoterpenes have the aldehyde
functionality, especially for α-pinene ozonolysis, in which all the primary and many later-generation
$RO_2$ contain at least one aldehyde group (Noziere et al., 2015; Berndt et al., 2018; Li et al., 2019;
Berndt, 2022; Zhao et al., 2022). As a result, acyl $RO_2$ may comprise a considerable fraction of total
$RO_2$ species and contribute significantly to the formation of low-volatility products and SOA in the
monoterpene oxidation system. A recent study by Zhao et al. (2022) found that the acyl $RO_2$-
involved reactions contribute to 50%-80% of oxygenated $C_{15}$-$C_{20}$ dimers (O:C $\geq$ 0.4) and 70% of
$C_{15}$-$C_{19}$ dimer esters in SOA from α-pinene ozonolysis. However, currently the direct probing of
the molecular identities and chemistry of monoterpene-derived acyl $RO_2$ radicals is rather limited.
The role of acyl $RO_2$ in HOM formation remains to be quantified.
In this study, the molecular identities and formation mechanisms of acyl $RO_2$ radicals, as well as
their contributions to HOM formation in the α-pinene ozonolysis are investigated. The experiments
were conducted in a flow reactor with different concentrations of $NO_2$, which acted as an efficient
scavenger for the acyl $RO_2$. The molecular composition and abundance of the gas-phase HOMs
were measured by a chemical ionization-atmospheric pressure interface-time-of-flight mass
spectrometer (CI-APi-TOF) using nitrate as the reagent ions. In addition, kinetic modelling using
the Framework for 0-D Atmospheric Modeling (F0AM v4.1) employing Master Chemical



Mechanisms (MCM v3.3.1) updated with the latest advances of the $RO_2$ chemistry was performed
to gain insights into the reaction kinetics and mechanisms of acyl $RO_2$ species. We find that acyl
$RO_2$ account for a major fraction of highly oxygenated $C_7$ and $C_8$ $RO_2$ and play a significant role in
the formation of HOM monomers and dimers with small molecular size. This study will help to
understand the role of acyl $RO_2$ in the formation of low-volatility species from monoterpene
oxidation and reduce the uncertainties in the future atmospheric modelling of the formation and
impacts of aerosols.
**2.    Method and Materials**
**2.1 Flow Reactor Experiments.**
The α-pinene ozonolysis experiments were carried out under room temperature (298 K) and dry
conditions (relative humidity < 5%) in a custom-built flow reactor, which has been described in detail
previously (Yao et al., 2019). The α-pinene vapor was generated by evaporating its pure liquid (99%,
Sigma-Aldrich) into a flow of zero air (10.65 L min$^{-1}$) added to the reactor using an automated
syringe pump (TYD01-01-CE, Baoding Leifu Fluid Technology Co., Ltd.). The initial
concentrations of α-pinene ranged from 500 ppb to 3 ppm in different experiments. Ozone was
generated by passing a flow of ultra-high-purity (UHP) $O_2$ (150 mL min$^{-1}$, Shanghai Maytor Special
Gas Co., Ltd.) through a quartz tube housing a pen-ray mercury lamp (UV-S2, UVP Inc.) and its
concentration (45 ppb and 180 ppb under low and high $O_3$ conditions, respectively) was measured
by an ozone analyzer (Model 49i, Thermo Fisher Scientific, USA). The $NO_2$, acting as an acyl $RO_2$
scavenger, was derived from its standard cylinder gas (15.6 ppm, Shanghai Weichuang Standard
Gas Co., Ltd.) and its initial concentration ranged from 0 to 30 ppb. To validate the formation
mechanisms of acyl $RO_2$, selected experiments with the addition of NO or cyclohexane were also
conducted. NO was derived by its standard cylinder gas (9.8 ppm, Shanghai Weichuang Standard
Gas Co., Ltd.) and its initial concentration also ranged from 0 to 30 ppb. The gas-phase cyclohexane
(~ 500 ppm), acting as an OH scavenger, was generated by bubbling a gentle flow of UHP $N_2$
through liquid cyclohexane (LC-MS grade, CNW). The total air flow in the flow reactor was 10.8 L
min$^{-1}$ and the residence time was 25 seconds. The relatively low $O_3$ concentration and short reaction
time in the flow reactor avoid significant production of $NO_3$ radicals from $NO_2$ and $O_3$ and make
the $NO_3$ oxidation contribute only 0.3%-1.2% of the total α-pinene oxidation in our experiments.
Therefore, the $NO_3$ chemistry could be neglected in this study. A summary of the experimental
conditions is given in Tables S1 and S2 in the Supplement.
The gas-phase $RO_2$ radicals and closed-shell products were measured by a nitrate-based CI-API-
TOF mass spectrometer (abbreviated as nitrate-CIMS; Aerodyne Research, Inc.), and a long time-



of-flight mass spectrometer with a mass resolution of ~10000 Th/Th was used here. The mass
calibration error is below 1.8 ppm. The sheath flow, including a 2 mL min⁻¹ UHP N₂ flow containing
nitric acid (HNO₃) and 22.4 L min⁻¹ zero air was guided through a PhotoIonizer X-ray (Model L9491,
Hamamatsu, Japan) to generate nitrate reagent ions. The total sample flow rate was 9 L min⁻¹ during
the experiments. The instrument was calibrated with a sulfuric acid (H₂SO₄) calibration factor and
a mass-dependent transmission efficiency. The mass spectra within the m/z range of 50 to 700 were
analyzed using the tofTools package developed by Junninen et al. (2010) based on Matlab. After
getting the signals of the gas-phase oxygenated organic molecules (OOMs), their concentration can
be calculated as follows (Jokinen et al., 2012; Bianchi et al., 2019):
$$[OOM] = C \times \frac{I_{OOM}}{I_{NO_3^-} + I_{HNO_3NO_3^-} + I_{HNO_3HNO_3NO_3^-}} \times \frac{1}{T_i} \quad (1)$$

$C$ is the calibration factor of H₂SO₄, with a value of $4.06 \times 10^9$ molecule cm⁻³ in this study; $I_X$ is the
detected signal of $X$ in the unit of counts per second (cps) and most OOMs were detected as adducts
with NO₃⁻; $T_i$ is the mass-dependent transmission efficiency of the instrument determined using the
following equation by adding propanoic acid, pentanonic acid and heptanonic acid vapors to deplete
NO₃⁻ (Figure S1):
$$T_i = 0.56 + 7.2 \times 10^4 / ((m/z - 498.84)^2 + 3.46 \times 10^4) \quad (2)$$

**2.2 Kinetic Model Simulations.**
Model simulations of RO₂ and HOM formation in selected experiments were performed to constrain
the reaction kinetics and mechanisms of acyl RO₂ using F0AM v4.1 (Wolfe et al., 2016), which
employs MCM v3.3.1 (Jenkin et al., 2015) updated with the chemistry of RO₂ autoxidation and
cross-reactions forming HOM monomers and dimers. Newly added species and reactions to MCM
v3.3.1 followed the work by Zhao et al. (2018) and Wang et al. (2021). Considering that the default
MCM v3.3.1 does not include highly oxygenated acyl RO₂, we added the possible formation
pathways of the potential acyl RO₂ measured in this study to the model based on the mechanisms
proposed by Zhao et al. (2022).
The formation and reaction branching ratios of the two α-pinene-derived CIs are updated in the
model according to the recent studies (Table S3) (Claflin et al., 2018; Iyer et al., 2021; Zhao et al.,
2021; Berndt, 2022). The formation of a ring-opened C₁₀H₁₅O₄-RO₂ species (C10H15O4RBRO2 in
Table S3) from α-pinene ozonolysis proposed by a recent study (Iyer et al., 2021), as well as its
subsequent autoxidation and bimolecular reactions, is included in the model. The autoxidation rate
constant of the ring-opened C₁₀H₁₅O₄-RO₂ is 1 s⁻¹, and a lower limit of its molar yield (30%) was




used according to the recent studies (Zhao et al., 2021; Meder et al., 2023) and our results (see
details in Section 3.3). We also added the hydrogen abstraction channel of α-pinene oxidation by
OH radicals according to a recent study (Shen et al., 2022). The branching ratio of this channel was
set to 9%, with the rest 91% being the traditional OH addition pathways. The detailed reaction
pathways and rate constants of $RO_2$ species in this channel followed the work by Shen et al. (2022),
except for $RO_2$ cross-reactions, the rates of which were not reported in that study. As the primary
$RO_2$ radicals ($C_{10}H_{15}O_2$-$RO_2$) formed via the hydrogen abstraction by OH radical are least-oxidized
with only 2 oxygen atoms, their cross-reaction rate could be relatively low (Atkinson et al., 2007;
Orlando and Tyndall, 2012). In the model, this rate constant was set to $1 \times 10^{-13}$ $cm^3$ $molecule^{-1}$ $s^{-1}$.
For other alkyl $RO_2$ radicals (including HOM-$RO_2$), their cross-reaction rate constant is assumed to
be $1 \times 10^{-12}$ $cm^3$ $molecule^{-1}$ $s^{-1}$ according to Zhao et al. (2018). The dimer formation rates for these
alkyl $RO_2$ are same as their cross-reaction rates.
In flow reactor experiments, the equilibrium formation of $ROONO_2$ would lead to the consumption
of alkyl $RO_2$ radicals. To account for the influence of this process on the $RO_2$ budget and HOM
formation, we included the reaction of $RO_2$ + $NO_2$ ⇋ $ROONO_2$ in the model, with forward and
reverse reaction rate constants of $7.5 \times 10^{-12}$ $cm^3$ $molecule^{-1}$ $s^{-1}$ and 5 $s^{-1}$, respectively (Orlando and
Tyndall, 2012). To simply the parameterization, the forward and reverse reaction rate constants of
newly added highly oxygenated acyl $RO_2$ with $NO_2$ are the same as default values in MCM v3.3.1.
Besides, the cross-reaction rate constants of acyl $RO_2$ (including acyl $RO_2$ + acyl $RO_2$ and acyl $RO_2$
+ alkyl $RO_2$) forming monomers or dimers were both set to $1 \times 10^{-11}$ $cm^3$ $molecule^{-1}$ $s^{-1}$ (Orlando
and Tyndall, 2012). Considering that there are large uncertainties in the dimer formation rate of $RO_2$,
a sensitivity analysis was conducted to evaluate its influence on acyl $RO_2$-involved HOM formation
by varying the rate constant from $1 \times 10^{-13}$ $cm^3$ $molecule^{-1}$ $s^{-1}$ to $1 \times 10^{-12}$ $cm^3$ $molecule^{-1}$ $s^{-1}$ for alkyl
$RO_2$ and $1 \times 10^{-12}$ $cm^3$ $molecule^{-1}$ $s^{-1}$ to $1 \times 10^{-11}$ $cm^3$ $molecule^{-1}$ $s^{-1}$ for acyl $RO_2$. The results show that
changes in dimer formation rate constants within the above ranges have no significant influence on
the contribution of acyl $RO_2$ to HOM formation (Figure S2).
The wall losses of OH, $HO_2$, and $RO_2$ radicals, as well as closed-shell HOM monomers and dimers
in the flow reactor were considered using the KPS method proposed by Knopf et al. (2015) in the
model (Table S4), with an assumption of irreversible uptake of these species on the reactor wall. It
is found that the wall loss of OH, $HO_2$, and $RO_2$ radicals accounts for 0.08-0.14%, 4.7-9.1%, and 7.3-
25.5% of their total production, respectively, with lower values under higher reacted α-pinene
concentration conditions. Therefore, the wall loss process would not significantly influence α-
pinene oxidation and $RO_2$ chemistry. The wall losses of closed-shell HOM monomers and dimers





account for 18.4-34.7% and 14.2-33.1% of their total production, respectively. It should be noted
that the wall losses of typical $RO_2$ and HOMs have negligible impact on their responses to the
addition of $NO_2$ (Figure S3). In addition, with the consideration of the wall loss effects, the effect
and contribution of acyl $RO_2$ to the HOM formation only changed a little (0.02-0.5%). Therefore,
the wall losses of $RO_2$ and HOMs in the flow reactor would not affect the interpretation of the results
in this study.
**3.    Results and Discussion**
**3.1 Molecular composition of acyl $RO_2$ from α-pinene ozonolysis**
The overall formation characteristics of gas-phase $RO_2$, closed-shell monomers, and dimers with
the addition of $NO_2$ (30 ppb) is shown in Figure 1 (Exps 8 and 14, Table S1). Since nitrate-CIMS is
only highly sensitive to the highly oxygenated species, we only discuss the production of HOMs
with oxygen atoms above 6 here. As for $RO_2$ and closed-shell monomers (Figure 1a), the
concentrations of $C_7$ and $C_8$ species decrease by more than 50% with the addition of $NO_2$, while for
$C_9$ and $C_{10}$ species, their decreases are relatively small (within 40%). In addition, we note that there
is an unexpected increase in some $C_9$ and $C_{10}$ $RO_2$, and the possible reason will be discussed in
detail in Section 3.3.

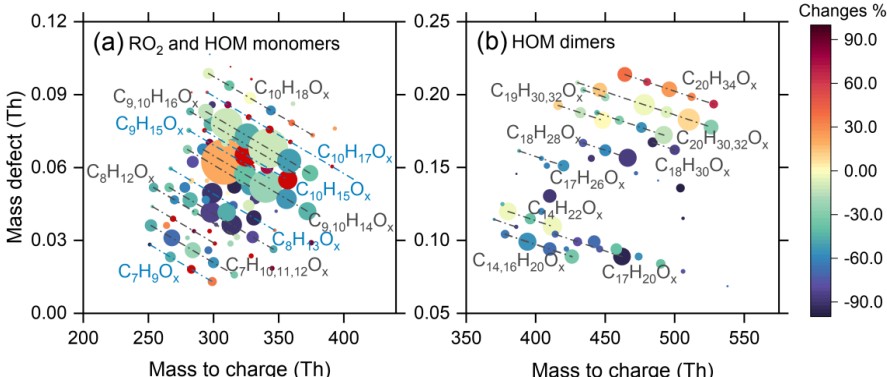


Figure 1 Mass defect plots of (a) $RO_2$, HOM monomers, and (b) HOM dimers formed from
ozonolysis of α-pinene in the presence of $NO_2$ measured using nitrate-CIMS (Exps 8, 14). The
circles are colored by the relative changes in concentration of $RO_2$, monomers and dimers due to
the addition of $NO_2$ (30 ppb). The area of circles is linearly scaled with the cube root of the
concentration of HOMs formed in the absence of $NO_2$. The blue lines represent $RO_2$ radicals.
$NO_2$ could react rapidly with acyl $RO_2$ radicals to form $RC(O)OONO_2$, which has a higher thermal-
stability compared to $ROONO_2$ and can serve as a sink for acyl $RO_2$ on our experimental timescales.



Therefore, a significant decrease in $C_7$ and $C_8$ $RO_2$ and HOMs upon the addition of $NO_2$ indicates
that a major fraction of $C_7$ and $C_8$ $RO_2$ are acyl $RO_2$. In contrast, the slight decrease in $C_9$ and $C_{10}$
HOM monomers shows that the contribution of acyl $RO_2$ to $C_9$ and $C_{10}$ $RO_2$ is relatively small.
However, some of the $C_{10}$ monomers showed a slight increase with the addition of $NO_2$, especially
for $C_{10}H_{18}O_x$-HOMs. The addition of $NO_2$ plays a twofold role in dimer formation from α-pinene
ozonolysis (Figure 1b). There is a significant inhibiting effect on $C_{14}$-$C_{18}$ dimers, which is due to
the large contribution of acyl $RO_2$ to the total $C_7$ and $C_8$ $RO_2$ that generate such dimers. However,
$C_{19}$ and $C_{20}$ dimers only show a slight decrease with the addition of $NO_2$, and some of them are even
enhanced. In particular, the enhancement in $C_{20}H_{34}O_x$ is most significant, reaching 30%.
Kinetic model simulations show that the concentration of alkyl $RO_2$ decreases by 1-20% with the
addition of 30 ppb $NO_2$ under different reacted α-pinene conditions (Exps 1-28). Considering that
the acyl $RO_2$ could be rapidly consumed by $NO_2$, if the concentration reduction of a $RO_2$ species
significantly exceeds 20% with 30 ppb $NO_2$ addition, we presume it has significant contribution
from acyl $RO_2$. As a result, a total of 10 acyl $RO_2$ were identified according to the changes of $RO_2$
concentration as a function of initial $NO_2$ concentration, which include $C_7H_9O_6$, $C_7H_9O_7$, $C_8H_{13}O_6$,
$C_8H_{13}O_8$, $C_8H_{13}O_9$, $C_8H_{13}O_{10}$, $C_9H_{13}O_9$, $C_9H_{17}O_7$, $C_9H_{17}O_9$, and $C_{10}H_{15}O_7$. Figure 2 shows the
averaged normalized acyl $RO_2$ concentrations measured as a function of the added $NO_2$
concentration under different experimental conditions (Exps 1-28). Similarly, since nitrate-CIMS is
only highly sensitive to products with high oxygen content, we only observed acyl $RO_2$ with oxygen
atoms above 6. Consistent with the significant decrease in $C_7$ and $C_8$ species with the addition of
$NO_2$ in Figure 1a, $C_7$ and $C_8$ acyl $RO_2$ decrease by more than 50% with the increase of $NO_2$
concentration (Figures 2a, b). For $C_9$ acyl $RO_2$, the $C_9H_{17}O_7$-$RO_2$ and $C_9H_{17}O_9$-$RO_2$ also decrease
dramatically with increasing $NO_2$, and the decrease in $C_9H_{13}O_9$-$RO_2$ is relatively smaller (Figure
2c). In addition, $C_{10}H_{15}O_7$-$RO_2$ also shows a small decrease (Figure 2d), with a reduction of only
30% at 30 ppb $NO_2$. The relative small reduction in the abundance of some of these $RO_2$ radicals
indicates the presence of alkyl $RO_2$ radicals with the same chemical formulas.

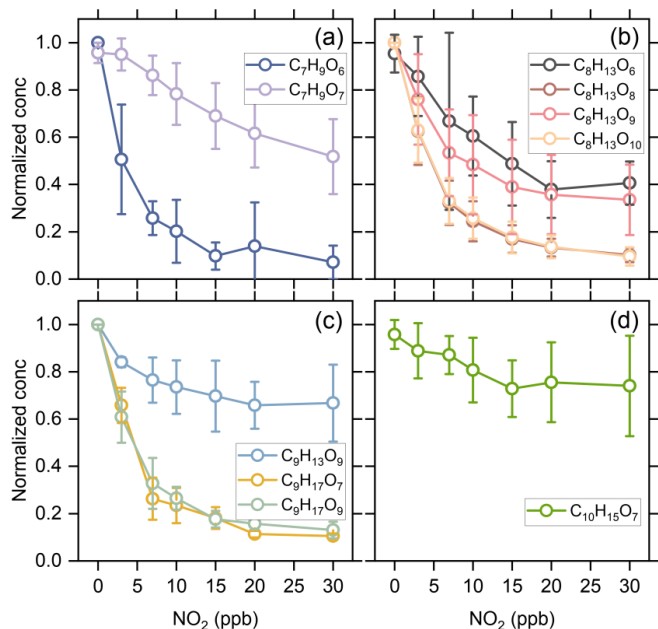

Figure 2 Averaged normalized concentration of the measured acyl $RO_2$ as a function of the added $NO_2$ concentration under different experimental conditions (Exps 1-28).

Figure 3 shows the contribution of acyl and alkyl $RO_2$ to the highly oxidized $C_7$-$C_{10}$ $RO_2$. Acyl $RO_2$ contribute 67.2%, 94.3% and 31.9% to the total $C_7$, $C_8$, and $C_9$ $RO_2$ concentrations, respectively. By contrast, the only $C_{10}$ acyl $RO_2$ measured in this study is $C_{10}H_{15}O_7$, which contributes to only 0.5% of the total $C_{10}$ $RO_2$. It should be note that there might be other $C_{10}$ acyl $RO_2$ that were not observed due to the interferences from the alkyl $RO_2$ with the same chemical formulas, which respond differently to the addition of $NO_2$ than acyl $RO_2$ do (see details in the following discussion). Considering that some $RO_2$ formulas such as $C_{10}H_{15}O_7$ may have contributions from both acyl $RO_2$ and alkyl $RO_2$, we assumed the decrease of $RO_2$ concentration with the addition of $NO_2$ as the concentration of acyl $RO_2$. Besides, it is obvious that the normalized concentration basically decreases to the lowest value when the initial $NO_2$ concentration reaches 10 ppb (Figure 2), indicating that most of the acyl $RO_2$ are depleted at this $NO_2$ concentration. In addition, the decreasing extents of some acyl $RO_2$ are different for different reacted α-pinene concentrations, with lower decreasing extent for higher reacted α-pinene concentrations (Figure S4). This difference might be due to the promoted cross-reactions of acyl $RO_2$ as well as their precursor $RO_2$ at higher α-pinene concentrations, which are competitive with the reactions leading to acyl $RO_2$ formation as well as the acyl $RO_2$ + $NO_2$ reactions.



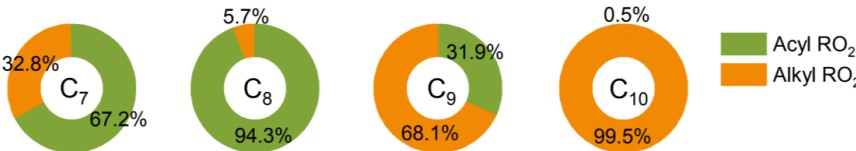

Figure 3 Contributions of acyl and alkyl $RO_2$ to the highly oxygenated $C_7$-$C_{10}$ $RO_2$ measured by nitrate-CIMS.

In addition to the changes of acyl $RO_2$ concentration, we also show the changes of normalized alkyl $RO_2$ concentration with the increasing initial $NO_2$ concentration in Figure S5. Although $ROONO_2$ formed by the reaction of alkyl $RO_2$ with $NO_2$ is thermally unstable and would decompose quickly to release $RO_2$, it would still reach a formation/decomposition equilibrium in the system, thus consuming a small amount of alkyl $RO_2$. However, it can be seen from Figure S5 that during 25 s of reaction in the flow reactor, a large part of alkyl $RO_2$ has an increasing trend with the increase of $NO_2$ concentration. We speculate that a portion of $ROONO_2$ could decompose back to $RO_2$ and $NO_2$ in the nitrate-CI inlet where the sample gases were diluted instantly and the equilibrium of $ROONO_2$ was disturbed, resulting in the release of a large amount of $RO_2$.

To verify our speculation, the decomposition of $ROONO_2$ in the CI inlet was simulated based on the dilution ratio (1:3.5) and residence time (200 ms) in the inlet. As shown in Figure S6, more than 40% of $ROONO_2$ decompose back to $RO_2$ and $NO_2$ in the CI inlet, which would inevitably lead to an increase in $RO_2$ concentration. As the $C_{10}H_{15}O_8NO_2$ has a significant contribution from the relative stable $RC(O)OONO_2$ arising from the ring-opened acyl $C_{10}H_{15}O_8$-$RO_2$ reported by Iyer et al. (2021), its decomposition is relatively small (~21%). It should be noted that the $RO_2$ measured here is only a part of total $RO_2$ and that a large amount of $RO_2$ has already reacted to form closed-shell products as well as $ROONO_2$ in the flow reactor. Taking Exp 14 as an example (30 ppb $NO_2$), the simulated concentrations of $RO_2$ and $ROONO_2$ are 1.3 ppb and 1.9 ppb, which approximately accounts for 27.1% and 39.6% of the total production of $RO_2$, respectively. Therefore, the decomposition of $ROONO_2$ could indeed result in an increase in the $RO_2$ concentration. It should also be pointed out that because of the very short residence time in the CI inlet, such an increase in the $RO_2$ concentration would not significantly impact HOM formation.

To confirm the reliability of our results, we examined the changes in the concentrations of $RO_2$ and closed-shell products as a function of reacted α-pinene in the absence of $NO_2$ (Section S1 and Figure S7), and the results are consistent with previous studies (Zhao et al., 2018). In addition, we repeated Exps 15-21 on another nitrate-CIMS and a similar increase in alkyl $RO_2$ signals with the addition of $NO_2$ was observed on that instrument (Figure S8).



**3.2 Formation mechanisms of acyl RO$_2$ during α-pinene ozonolysis**

It has been recently suggested that there are three main pathways that directly lead to the formation of monoterpene-derived acyl RO$_2$ (Zhao et al., 2022): (i) the autoxidation of RO$_2$ containing aldehyde groups (Reaction R1), (ii) the cleavage of C-C bond of RO containing an α-ketone group (Reaction R2), and (iii) the intramolecular H-shift of RO containing an aldehyde group (Reaction R3). Here, we further investigated the formation mechanisms of acyl RO$_2$. Figure 4 shows the reaction schemes leading to the formation of example acyl RO$_2$ radicals. The detailed formation mechanisms of acyl RO$_2$ measured in this study are shown in Figure S9. The formation of acyl RO$_2$, especially those having the small molecular size (C$_7$-C$_9$), requires the production and subsequent decomposition (or ring-opening process) of RO radicals. Take C$_8$H$_{13}$O$_6$-RO$_2$ as an example (Figure 4), two steps of RO formation and decomposition following the primary C$_{10}$H$_{15}$O$_4$-RO$_2$ lead to the ring-opened C$_8$H$_{13}$O$_4$-RO$_2$ that can undergo rapid aldehydic H-shift to form the acyl RO$_2$. While for C$_8$H$_{13}$O$_9$-RO$_2$, it directly comes from the aldehydic H-shift of C$_8$H$_{13}$O$_7$-RO followed by the O$_2$ addition (Figure S9).

$$RO_2 \xrightarrow{autox} acyl\ RO_2 \quad (R1)$$

$$RO_2 \xrightarrow{+RO_2} RO \xrightarrow{C-C\ cleavage} acyl\ RO_2 \quad (R2)$$

$$RO_2 \xrightarrow{+RO_2} RO \xrightarrow{H\ shift} acyl\ RO_2 \quad (R3)$$



315

Figure 4 Three different formation pathways of acyl $RO_2$ during ozonolysis of α-pinene. The acyl $RO_2$, $C_9H_{15}O_4$ and $C_9H_{15}O_5$, formed via pathways 2 and 3, respectively, were not detected by nitrate-CIMS in this study due to their relatively low oxygenation level.

To verify the formation mechanisms of acyl $RO_2$, we added NO in some experiments (Exps 33-56) to see how acyl $RO_2$ respond to the increasing NO concentration. As shown in Figure 5, the changes of $C_7$ and $C_8$ acyl $RO_2$ show opposite trend with the increasing NO and $NO_2$ concentration, except for $C_8H_{13}O_8$-$RO_2$. NO can react with $RO_2$ to form RO radicals and promote the formation of $RO_2$ that requires the involvement of RO radicals in their formation. In addition to $C_8H_{13}O_6$-$RO_2$ discussed above, the formation of $C_7H_9O_7$-$RO_2$ and $C_8H_{13}O_9$-$RO_2$ needs 2 and 4 steps of the RO formation following $C_{10}H_{15}O_4$-$RO_2$ (Figure S9), respectively. Therefore, the increase of RO concentration due to the addition of NO would promote the production of these acyl $RO_2$. These results prove that the RO radicals indeed play an important role in the acyl $RO_2$ formation. While for $C_8H_{13}O_8$-$RO_2$, its concentration decreases substantially with the addition of NO up to 3 ppb, similar to the trend observed with the addition of $NO_2$. After reaching the minimum at 7 ppb NO, the concentration of $C_8H_{13}O_8$-$RO_2$ tends to increase with the further increase of NO concentration. Given that $C_8H_{13}O_8$-$RO_2$ is likely to directly come from the autoxidation of $C_8H_{13}O_6$ acyl $RO_2$ (see Figure S9), the rapid consumption of $C_8H_{13}O_6$-$RO_2$ by NO and $NO_2$ (formed by $O_3$ oxidation of NO) may outcompete its autoxidation process, thus leading to a decrease in $C_8H_{13}O_8$-$RO_2$ concentration. Besides, it can be seen that the increasing extent in $C_8H_{13}O_6$-$RO_2$ is also relatively small before the NO concentration reaches 3 ppb (Figure 5c), indicating that the promotion effect of NO on $C_8H_{13}O_6$-



$RO_2$ formation is not that strong at this concentration.

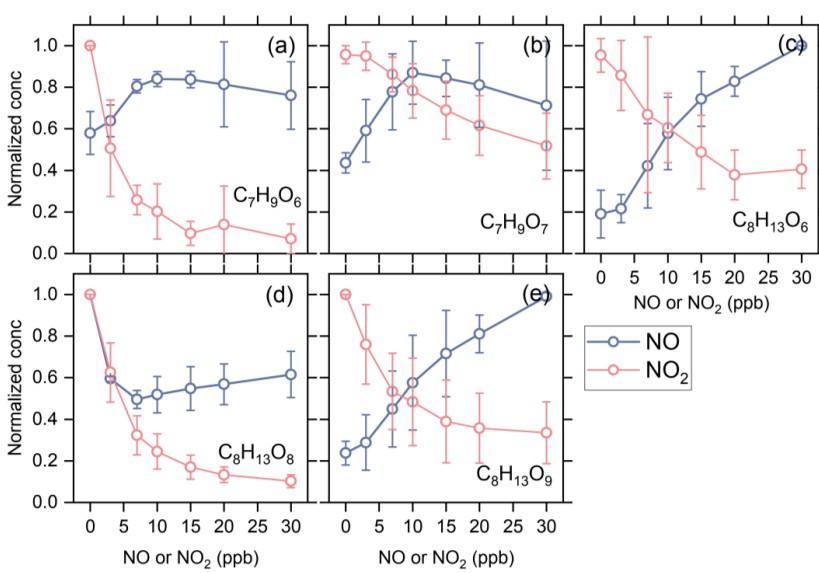


Figure 5 Averaged normalized concentration of typical acyl $RO_2$ as a function of initial NO or $NO_2$
addition (Exps 1-28 and 33-56).
It is interesting to note that most of the measured highly oxygenated acyl $RO_2$ are formed by the
autoxidation of aldehydic $RO_2$, and only the $C_8H_{13}O_9$-$RO_2$ is formed by the H-shift of the RO radical
(Figure S9). The concentration of acyl $RO_2$ from the autoxidation pathway accounts for 96% of all
highly oxygenated acyl $RO_2$ concentrations. Considering that the acyl $RO_2$ with small molecular
size are generally the ring-opened $RO_2$, the autoxidation rate constant of their precursor $RO_2$ is
expected to be relatively high (e.g., 1 s$^{-1}$) (Iyer et al., 2021). Taking a $RO_2$ cross-reaction rate
constant of $1 \times 10^{-12}$ cm$^3$ molecule$^{-1}$ s$^{-1}$ (Zhao et al., 2018) and a model-predicted total $RO_2$
concentration of 1.7 ppb (Exp 8), autoxidation and cross-reactions contribute to 96.0% and 4.0% of
the total $RO_2$ reaction, respectively. Considering a 10 times larger $RO_2$ cross-reaction rate constant
(i.e., $1 \times 10^{-11}$ cm$^3$ molecule$^{-1}$ s$^{-1}$), the contributions of $RO_2$ autoxidation and cross-reactions would
be 70.4% and 29.6%, respectively. These calculations suggest that the autoxidation of aldehydic
$RO_2$ plays a dominant role in the formation of the highly oxygenated acyl $RO_2$. Although the acyl
$RO_2$ with low oxygen content were not measured in this study, all acyl $RO_2$ containing oxygen atoms
less than 6 seem to be derived from the cleavage of C-C bond or H-shift of RO containing an α-
ketone or aldehyde in the currently known reaction mechanisms (Figures 4 and S10).
Recently, Shen et al. (2022) found that the hydrogen abstraction by OH radicals during α-pinene
oxidation plays an important role in HOM formation. In such mechanisms, the primary $RO_2$ reacts



with NO and forms RO radicals, which could undergo rapid ring-breaking reactions to form a series
of ring-opened $C_{10}H_{15}O_x$-$RO_2$, which contains aldehyde functionality and can easily autoxidize to
$C_{10}$ acyl $RO_2$. In the absence of NO, the cross-reactions of $RO_2$ can also produce RO radicals.
However, only a few $C_{10}$ acyl $RO_2$ were detected in this study and they contribute less than 1% of
the total $C_{10}$ $RO_2$ concentration. This phenomenon could be due to the fact that the primary $RO_2$
($C_{10}H_{15}O_2$) formed by the hydrogen abstraction by OH radical are least-oxidized with only 2 oxygen
atoms, which are expected to have a relatively low cross-reaction rate constant (Orlando and Tyndall,
2012; Berndt et al., 2018). As a result, the formation of ring-opened $C_{10}H_{15}O_x$-$RO_2$ via cross-
reactions of the primary $C_{10}H_{15}O_2$-$RO_2$ may not be important. As shown in Figure 6, when the cross-
reaction rate constants of $C_{10}H_{15}O_2$-$RO_2$ is considered to be $1\times10^{-13}$ $cm^3$ molecule$^{-1}$ s$^{-1}$, the simulated
contribution of the H-abstraction pathway to the HOM formation is less than 3% under both low
(2.4 ppb) and high (9.6 ppb) reacted α-pinene conditions. It should be note that the cross-reaction
rate constants of the less-oxygenated $RO_2$ could be even lower (Orlando and Tyndall, 2012),
therefore the contribution of this pathway to HOM formation could be ignored when NO is absent.

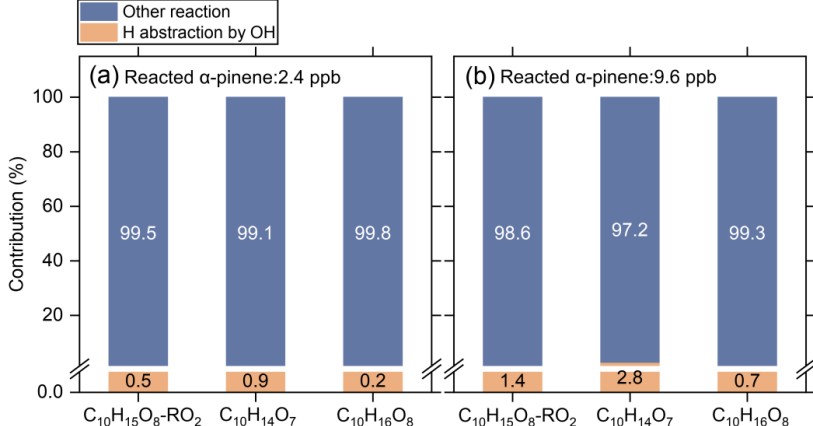


Figure 6 Contributions of the H-abstraction pathways by OH radicals (yellow) and OH addition and
ozonolysis pathways (blue) to the formation of typical HOMs under low (a) and high (b) reacted α-
pinene conditions simulated by the kinetic model. The cross-reaction rate constant was set to $1\times10^{-13}$
$cm^3$ molecule$^{-1}$ s$^{-1}$ for the primary $C_{10}H_{15}O_2$-$RO_2$ and $1\times10^{-12}$ $cm^3$ molecule$^{-1}$ s$^{-1}$ for the more
oxygenated $RO_2$.
In the presence of cyclohexane as an OH scavenger (Figure S11, Exp 32), the concentrations of
$C_{10}H_{17}O_x$-$RO_2$ formed via OH addition channel and the corresponding $C_{10}H_{18}O_x$-HOMs decrease
by more than 70%, while the $C_{10}H_{15}O_x$-$RO_2$ and its related closed-shell products decrease by less
than 15%, in good agreement with the measurements in previous studies (Zhao et al., 2018). As the



$C_{10}H_{16}O_8$-HOM could come from both $C_{10}H_{15}O_x$-$RO_2$ and $C_{10}H_{17}O_x$-$RO_2$, its reduction is at a
medium level. The significantly smaller decrease in the concentrations of $C_{10}H_{15}O_x$-$RO_2$ and its
corresponding closed-shell products as compared to those of $C_{10}H_{17}O_x$-$RO_2$ and the related closed-
shell products further illustrates that the H-abstraction by OH has a minor contribution to HOM
formation in the absence of NO.

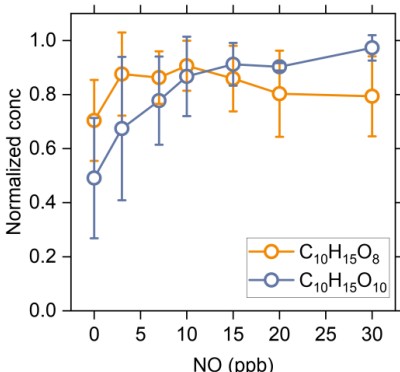


Figure 7 Averaged normalized concentration of the measured $C_{10}H_{15}O_8$- and $C_{10}H_{15}O_{10}$-$RO_2$ as a
function of the added NO concentration (Exps 33-56).
Figure 7 shows the changes in measured concentration of $C_{10}H_{15}O_8$-$RO_2$ and $C_{10}H_{15}O_{10}$-$RO_2$ as a
function of initial NO concentration (Exps 33-56). It should be noted that due to the existence of $O_3$
in our experiments, these two $RO_2$ could come from both $O_3$ and OH reactions with α-pinene and
NO could be rapidly oxidized to $NO_2$ by $O_3$. The normalized concentrations of $C_{10}H_{15}O_8$-$RO_2$ and
$C_{10}H_{15}O_{10}$-$RO_2$ increase firstly under low NO conditions, which is similar to the change of acyl $RO_2$
as shown in Figure 5. This increase could be due to two reasons: (1) the promoted formation of
$C_{10}H_{15}O_8$ and $C_{10}H_{15}O_{10}$ acyl $RO_2$ from the H-abstraction channel by NO addition and (2) the
equilibrium decomposition of $ROONO_2$ formed by the two alkyl $RO_2$ from ozonolysis of α-pinene
in the CI inlet (see Section 3.1). As mentioned above, the ring-opened $C_{10}H_{15}O_x$-$RO_2$ formed from
the H-abstraction channel contain aldehyde functionality and can autoxidize rapidly. The F0AM
model simulations show that the $C_{10}H_{15}O_8$ and $C_{10}H_{15}O_{10}$ acyl $RO_2$ formed from the H-abstraction
channel contribute to 68% and 56% of the total $C_{10}H_{15}O_8$-$RO_2$ and $C_{10}H_{15}O_{10}$-$RO_2$ with the addition
of 10 ppb NO, respectively. Therefore, the initial increases of these two $RO_2$ with increasing NO
concentration are likely mainly due to the enhanced formation of $C_{10}H_{15}O_8$ and $C_{10}H_{15}O_{10}$ acyl $RO_2$.
When the NO concentration increases to a high level, there are more NO and $NO_2$ in the system,
which promotes the consumption of acyl $RO_2$. As a result, $C_{10}H_{15}O_8$-$RO_2$ exhibits a decreasing trend
and the increasing extend of $C_{10}H_{15}O_{10}$-$RO_2$ becomes much smaller.




### 3.3 Contributions of acyl RO$_2$ to the formation of gas-phase HOMs


With the addition of NO$_2$, the distribution of gas-phase products in the α-pinene ozonolysis changes
significantly (see Figure 1), and the consumption of acyl RO$_2$ by NO$_2$ plays an important role. NO$_2$
influences the formation of HOM monomers mainly in three ways. Firstly, NO$_2$ could react rapidly
with acyl RO$_2$ and form RC(O)OONO$_2$, thus inhibiting the formation of HOMs with the
involvement of acyl RO$_2$. Secondly, as mentioned above, although ROONO$_2$ is thermally unstable,
their formation/decomposition equilibrium still consumes a small amount of alkyl RO$_2$, resulting in
a decrease in HOM formation. Thirdly, NO$_2$ can consume a part of HO$_2$ radicals (Figure S12, thus
inhibiting the RO$_2$ + HO$_2$ reaction pathway.

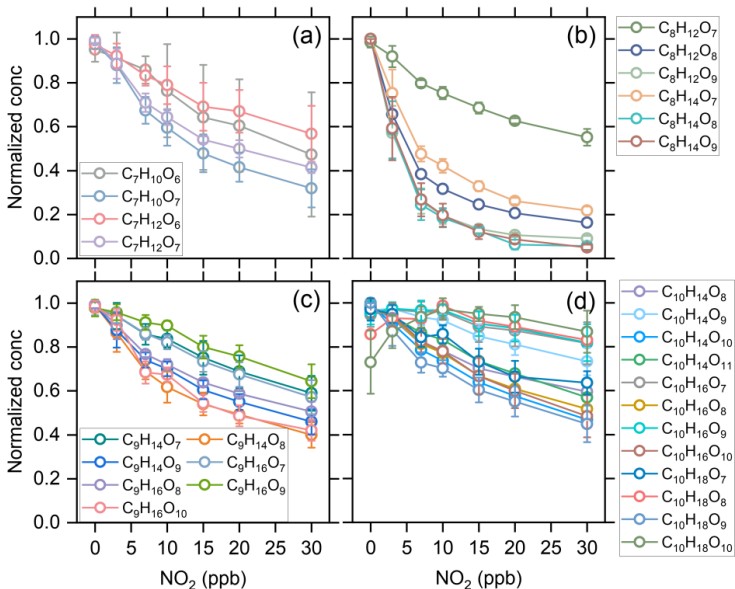


Figure 8 Averaged normalized concentration of the measured C$_7$-C$_{10}$ HOMs as a function of the
added NO$_2$ concentration (Exps 1-28).
Figure 8 shows the normalized concentration of C$_7$-C$_{10}$ HOM monomers as a function of initial NO$_2$
concentration. The C$_7$, C$_8$, and some of C$_9$ HOMs decrease significantly with increasing NO$_2$
concentration due to the relatively large contribution of acyl RO$_2$ to the total C$_7$-C$_9$ RO$_2$. The C$_7$
HOMs decrease by more than 50% when the NO$_2$ concentration reaches 30 ppb, while C$_8$ HOMs
decrease by more than 70% and some of them even decrease by 90%. The C$_9$ HOMs decrease by
30%-60% and the species with relatively large decrease are mostly acyl RO$_2$-related HOMs. For
C$_{10}$ HOMs, although there is also an obvious decrease in their formation with the addition of NO$_2$,
most of them have a smaller decreasing extent compared to the C$_7$-C$_9$ HOMs due to the low





contribution of acyl $RO_2$ to the $C_{10}$ $RO_2$. It is worth noting that a few $C_{10}$ HOMs increase initially
with the addition of $NO_2$ up to 10 ppb, suggesting that there might be some processes that promote
the formation of their precursor $RO_2$ radicals and thus offset the inhibiting effect of $NO_2$.
As mentioned above, the addition of $NO_2$ has the most significant influence on the formation of
small HOM monomers. Combined with the large contribution (67-94%) of acyl $RO_2$ to the total $C_7$
and $C_8$ $RO_2$ (Figure 3), it can be considered that the reduction in the formation of $C_7$ and $C_8$ HOM
monomers with $NO_2$ addition is overwhelmingly due to the consumption of acyl $RO_2$ by $NO_2$. As a
result, acyl $RO_2$ was found to have a contribution of 50-90% to $C_7$ and $C_8$ HOM monomer formation
during α-pinene ozonolysis. Since acyl $RO_2$ also have a considerable contribution (32%) to the total
$C_9$ $RO_2$, an upper limit (30%-60%) of its contribution to $C_9$ HOMs could be derived with the
assumption that the decrease of $C_9$ HOMs with the addition of $NO_2$ is also mainly due to the
consumption of $C_9$-acyl $RO_2$ by $NO_2$. By contrast, acyl $RO_2$ account for a very small fraction (0.5%)
of the total $C_{10}$ $RO_2$, and their contribution to $C_{10}$ HOMs cannot be quantified based solely on the
experimental measurements given that the equilibrium reaction between alkyl $RO_2$ and $NO_2$ can
also affect the formation of HOMs. Therefore, we used the F0AM model to simulate the contribution
of acyl $RO_2$ to $C_{10}$ HOM formation according to the acyl $RO_2$ measured in this study and displayed
the results in Figure 9. It should be noted that the HOMs from the acyl $RO_2$ and its subsequent $RO_2$
(formed from acyl $RO_2$ reactions) are all considered as acyl $RO_2$-related HOMs in the model.
As mentioned above, the formation of ring-opened $C_{10}H_{15}O_4$-$RO_2$ reported by Iyer et al. (2021) is
included in the model, and its autoxidation produces a ring-opened acyl $C_{10}H_{15}O_8$-$RO_2$. When we
considered the upper limit of the yield of ring-opened $C_{10}H_{15}O_4$-$RO_2$ (89%) in the model and
assumes that the other primary $RO_2$ with the cyclobutyl ring autoxidize at a very slow rate (0.01 s$^{-1}$
), the simulated acyl $C_{10}H_{15}O_8$-$RO_2$ would contribute to ~80% of the total $C_{10}$ $RO_2$. However, we
could not see a large decrease in the measured concentration of $C_{10}H_{15}O_8$-$RO_2$ and its related HOM
monomers with the addition of $NO_2$. Similarly, a recent study by Zhao et al. (2022) found that the
$C_{10}H_{15}O_8$-related monomers and dimers in α-pinene SOA also did not show significant decreases
with $NO_2$ addition. There might be two reasons for the discrepancy between the simulations and
measurements. Firstly, the yield of the ring-opened $C_{10}H_{15}O_4$-$RO_2$ might be significantly smaller
than 89% (Zhao et al., 2021; Meder et al., 2023). Secondly, the autoxidation rate of other primary
$C_{10}H_{15}O_4$-$RO_2$ with the cyclobutyl ring could be significantly larger than 0.01 s$^{-1}$. Therefore, we
updated the branching ratios and autoxidation rates of the primary $RO_2$ during the α-pinene
ozonolysis in the model according to the recent studies (Kurten et al., 2015; Claflin et al., 2018;
Zhao et al., 2021; Berndt, 2022) (Table S3), and a lower limit (30%) of the ring-opened $C_{10}H_{15}O_4$-





$RO_2$ yield reported by Iyer et al. (2021) was used here. The simulated acyl $RO_2$-related HOMs
contribute to 14% of the total $C_{10}$ HOMs, which is slightly smaller than the measured decrease in
$C_{10}$ HOMs with the addition of $NO_2$. This discrepancy could be due to two reasons. Firstly, the
decrease in HOMs can partly result from the consumption of alkyl $RO_2$ and $HO_2$ radicals by the
addition of $NO_2$. Secondly, as mentioned above, there might be other $C_{10}$ acyl $RO_2$ that were not
observed in this study due to the decomposition of the $ROONO_2$ from the alkyl $RO_2$ with the same
formulas.
The contributions of acyl $RO_2$ to the formation of $C_7$-$C_9$ HOMs were also simulated (Figure 9). For
$C_7$ and $C_8$ HOMs, the model predicts a contribution of 52%-98% from acyl $RO_2$, which is consistent
with the measurements (50%-90%). However, the simulated contribution of acyl $RO_2$ to $C_9$ HOMs
is over 99%, which is not consistent with the measurements (Figure 8c). Recent studies indicated
that the CI radicals from α-pinene ozonolysis could not form the alkyl $C_9H_{15}O_3$-$RO_2$ (C96O2 in
default MCM v3.3.1) (Kurten et al., 2015; Zhao et al., 2021; Berndt, 2022). As a result, this primary
$C_9$ alkyl $RO_2$ was not considered in the model, and most of $C_9$ $RO_2$ considered are acyl $RO_2$ or from
acyl $RO_2$ reactions. In view of the significantly lower measured (less than 30-60%) than simulated
(over 99%) contribution of acyl $RO_2$ to $C_9$ HOMs, we speculate that a small part of CI radicals
might be able to form the $C_9H_{15}O_3$-$RO_2$, which could further react to form highly oxygenated alkyl
$C_9$ $RO_2$.

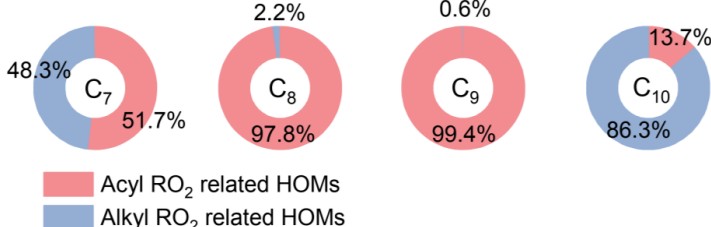


Figure 9 Simulated average contribution of acyl and alkyl $RO_2$ to $C_7$-$C_{10}$ HOM formation from
ozonolysis of α-pinene under typical experimental conditions (Exps 1, 8, 15, and 22).
A sensitive analysis of the alkyl $C_9H_{15}O_3$-$RO_2$ yield was conducted to see its influence on the
contribution of acyl $RO_2$ to the total $C_9$ HOMs. The model simulations show that when the yield of
this $C_9$ $RO_2$ from one of the CIs ranges between 0.5% to 2%, the contribution of acyl $RO_2$ to the
total $C_9$ HOMs ranges from 27.5% to 59.8% (Figure S13), which is almost consistent with the
measurements. This result indicates that a small part of CIs could generate the $C_9$ alkyl $RO_2$.
The cross-reaction rate constant of acyl $RO_2$ is generally larger than that of alkyl $RO_2$ (Atkinson et



al., 2007; Orlando and Tyndall, 2012), and the fast cross-reaction may lead to an important
contribution to the HOM dimer production. The responses of dimer formation to increasing
concentration of initial $NO_2$ during α-pinene ozonolysis are given in Figure 10. The $C_{14}$-$C_{18}$ dimers
decrease by up to 50%-95% with the increase of $NO_2$ concentration up to 30 ppb (Figures 10a-e).
The rapid cross-reaction rate of acyl $RO_2$, as well as their dominant contribution to the small $RO_2$
species makes acyl $RO_2$ an important contributor to the formation of these dimers. The consumption
of acyl $RO_2$ by $NO_2$ greatly inhibits the bimolecular reactions involving acyl $RO_2$, resulting in a
rapid decrease in the concentration of the corresponding dimers. Considering the predominance of
acyl $RO_2$ in small $RO_2$ and their high reaction rate with $NO_2$ compared to the alkyl $RO_2$, it can be
concluded that the cross-reactions involving acyl $RO_2$ contribute to roughly 50%-95% of the $C_{14}$-
$C_{18}$ dimer formation.
For $C_{19}$ dimers, due to the relatively smaller contribution of acyl $RO_2$ to $C_9$ and $C_{10}$ $RO_2$, their
concentration decreases only by 10%-40%, and this reduction have contributions from both acyl
and alkyl $RO_2$. For $C_{20}$ dimers, their concentration changes with the addition of $NO_2$ can be
discussed according to the number of hydrogen atoms in the molecules. Firstly, the concentration
of $C_{20}H_{30}O_7$ and $C_{20}H_{30}O_9$ decreases by 40-60% with the addition of 30 ppb $NO_2$, indicating a
significant contribution of acyl $RO_2$ such as $C_{10}H_{15}O_5$-$RO_2$ (acyl $RO_2$ in default MCM v3.3.1) and
$C_{10}H_{15}O_7$-$RO_2$ in their formation, while other $C_{20}H_{30}O_x$ dimers decrease by ~30%. The $C_{20}H_{32}O_x$
dimer series also exhibits a small reduction (less than 20%) with the addition of $NO_2$. However, the
$C_{20}H_{34}O_x$ series shows an unexpected increase with the addition of $NO_2$ up to 10 ppb and almost
remains unchanged with the further increase of $NO_2$ concentration. Given that the cross-reaction
rate constant of acyl $RO_2$ can be orders of magnitude higher than that of counterpart alkyl $RO_2$
(Atkinson et al., 2007; Orlando and Tyndall, 2012), the rapid consumption of acyl $RO_2$ by $NO_2$
would preserve the alkyl $RO_2$ that tend to react with acyl $RO_2$ at a fast rate in the absence of $NO_2$,
which to some extent would elevate the concentration of alkyl $RO_2$ in the system and thus promote
the less competitive alkyl $RO_2$ + alkyl $RO_2$ reactions to form $C_{20}H_{34}O_x$ dimers. The slight increase
of some $C_{10}H_{18}O_x$-HOMs with the addition of $NO_2$ up to 10 ppb could also be due to this reason.



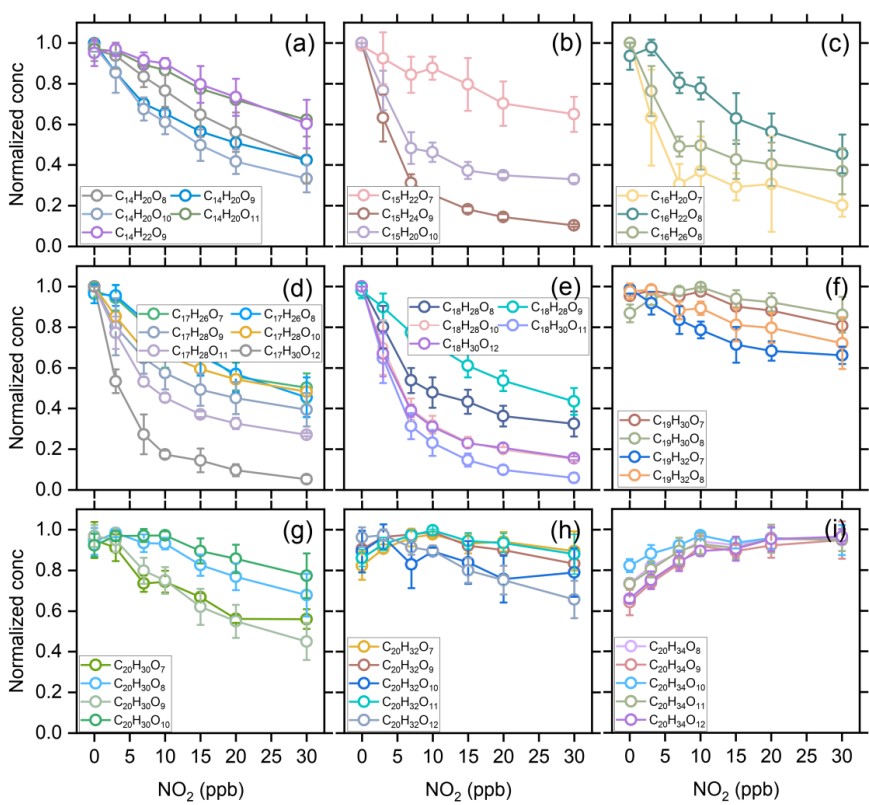

Figure 10 Averaged normalized concentration of the measured $C_{14}$-$C_{20}$ dimers as a function of the added $NO_2$ concentration (Exps 1-28).

According to the noticeable increasing trend in $C_{20}H_{34}O_x$ as compared to other $C_{20}$ dimers, we speculate that acyl $RO_2$ react faster with $C_{10}H_{17}O_x$ alkyl $RO_2$ than with $C_{10}H_{15}O_x$ alkyl $RO_2$. Therefore, when the acyl $RO_2$ is depleted, the preservation of $C_{10}H_{17}O_x$-$RO_2$ is more significant and the promotion of their cross-reactions to form $C_{20}H_{34}O_x$ is more evident. It is also possible that the reaction of $NO_2$ with $C_{10}H_{17}O_x$ alkyl $RO_2$ is less efficient compared to the reaction with $C_{10}H_{15}O_x$ alkyl $RO_2$, so more $C_{10}H_{17}O_x$ than $C_{10}H_{15}O_x$ are available for dimer formation in the presence of $NO_2$.

To further prove the above two speculations, we performed sensitivity analyses for the reaction rates of $C_{10}H_{17}O_x$-$RO_2$ using the F0AM model. Figures 11a show the changes in $C_{20}H_{34}O_x$ dimers with $NO_2$ addition at different $C_{10}H_{17}O_x$-$RO_2$ + $NO_2$ reaction rates under the conditions of Exps 8-14. As the reaction rate varies from $5\times10^{-14}$ to $1\times10^{-12}$ cm$^3$ molecule$^{-1}$ s$^{-1}$, the increasing trend of $C_{20}H_{34}O_x$ dimers versus the added $NO_2$ concentration is significantly weakened and the simulations are more




deviated from the measurements. When the reaction rate increases to $7.5\times10^{-12}$ cm$^3$ molecule$^{-1}$ s$^{-1}$,
the C$_{20}$H$_{34}$O$_x$ dimers decrease significantly with increasing NO$_2$, which is in striking contrast to the
measurements. Figure 11b presents the sensitivity analysis results for the cross-reaction rate
constants of acyl RO$_2$ + C$_{10}$H$_{17}$O$_x$-RO$_2$. As this rate constant varies from $1\times10^{-12}$ to $1\times10^{-10}$ cm$^3$
molecule$^{-1}$ s$^{-1}$, the increasing trend of C$_{20}$H$_{34}$O$_x$ versus the NO$_2$ concentration is more pronounced
and more consistent with the measurements. These sensitivity analyses support our speculation that
the C$_{10}$H$_{17}$O$_x$ alkyl RO$_2$ may be different from other alkyl RO$_2$ radicals in terms of the reaction
efficiency with NO$_2$ and acyl RO$_2$ species, which leads to different responses of C$_{20}$H$_{34}$O$_x$ dimers
to NO$_2$ addition compared to other C$_{20}$ dimers. These results also suggest that the presence of acyl
RO$_2$ could affect the fate and contribution of alkyl RO$_2$ to HOM formation in atmospheric oxidation
systems given the different reactivity of acyl RO$_2$ from alkyl RO$_2$.

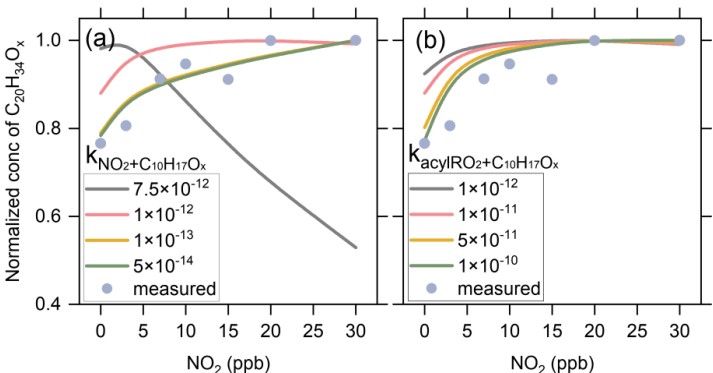


Figure 11 Sensitivity analyses of the reaction rates of NO$_2$ with C$_{10}$H$_{17}$O$_x$-RO$_2$ (a), and the cross-
reaction rate of acyl RO$_2$ with C$_{10}$H$_{17}$O$_x$-RO$_2$ (b), considering a rate of $1\times10^{-12}$ cm$^3$ molecule$^{-1}$ s$^{-1}$
for C$_{10}$H$_{17}$O$_x$-RO$_2$ + NO$_2$.
**4. Conclusions**
In this study, the molecular identities, formation mechanisms, and contributions of acyl RO$_2$ to the
formation of HOMs during ozonolysis of α-pinene are investigated using a combination of flow
reactor experiments and detailed kinetic model simulations. Based on the marked decrease in RO$_2$
concentration as a function of initial NO$_2$ concentration, a total of 10 acyl RO$_2$ are identified during
α-pinene ozonolysis. The acyl RO$_2$ contributes to 67%, 94% and 32% of C$_7$, C$_8$ and C$_9$ highly
oxygenated RO$_2$ but only 0.5% of C$_{10}$ highly oxygenated RO$_2$, respectively. Three main pathways
are identified for the formation of monoterpene-derived acyl RO$_2$: (i) the autoxidation of RO$_2$
containing aldehyde groups, (ii) the cleavage of C-C bond of RO containing an α-ketone group, and
(iii) the intramolecular H-shift of RO containing an aldehyde group. The autoxidation of aldehydic



RO$_2$ formed involving multiple RO decomposition or ring-opening steps plays a dominant role in
the formation of the highly oxygenated acyl RO$_2$ radicals (oxygen atom number $\geqslant$ 6), while the
less-oxygenated acyl RO$_2$ (oxygen atom number < 6) are mainly derived from the other two
pathways.
The acyl RO$_2$-involved reactions explain 50-90% of C$_7$ and C$_8$ HOM monomers and 14% of C$_{10}$
HOMs, respectively. For C$_9$ HOMs, this contribution can be up to 30%-60%. For the HOM dimers,
acyl RO$_2$-involved reactions contribute 50%-95% to the formation of C$_{14}$-C$_{18}$ dimers. Owing to the
higher cross-reaction rate constant of acyl RO$_2$ compared to alkyl RO$_2$, the acyl RO$_2$ + alkyl RO$_2$
reaction would outcompete the alkyl RO$_2$ + alkyl RO$_2$ reaction. Therefore, the rapid consumption
of acyl RO$_2$ by NO$_2$ in the experiments (as well as in polluted atmospheres) would make the alkyl
RO$_2$ that are supposed to react with acyl RO$_2$ retained, which to some extent elevates the
concentration of alkyl RO$_2$ in the system and thus promotes the reaction of alkyl RO$_2$ + alkyl RO$_2$
to form dimers such as C$_{20}$H$_{34}$O$_x$. The contribution of H-abstraction of α-pinene by OH radical to
the formation of acyl RO$_2$ and HOMs is found to be negligible in the absence of NO. This is because
the primary C$_{10}$H$_{15}$O$_2$-RO$_2$ radicals formed in such pathways are least-oxidized and thus have
relatively low cross-reaction efficiency to produce RO radicals, which are the key intermediates for
the formation of acyl RO$_2$ and HOMs in that channel. However, in the presence of NO, the formation
of highly oxygenated acyl RO$_2$ via the H-abstraction pathway is demonstrated, consistent with
previous studies (Shen et al., 2022).
In this study, acyl RO$_2$ species are identified according to a dramatic decrease in their concentration
with the addition of NO$_2$. It should be noted that the presence of NO$_2$ could also inhibit the formation
of alkyl RO$_2$ species involving acyl RO$_2$ reactions. If there are any contributions of alkyl RO$_2$ to
acyl RO$_2$ identified in this study, the influence of such alkyl RO$_2$ species on HOM formation would
reflect an indirect effect of acyl RO$_2$. However, given that the formation of most of the acyl RO$_2$
identified in this study can be reasonably explained by the proposed mechanisms and verified by
their responses to the addition of NO, the acyl RO$_2$ identified here are expected to have no significant
contributions from alkyl RO$_2$. Currently, the reaction kinetics of monoterpene-derived acyl RO$_2$ are
still poorly understood. Considering the important contribution of acyl RO$_2$ to HOM formation,
further kinetic studies are needed to get more specific rate constants for their autoxidation and cross-
reactions, thereby deepening our understanding of the role of acyl RO$_2$ in HOM and SOA formation
under atmospheric conditions.

*Data availability.* The data presented in this work are available upon request from the corresponding
author.




*Author contributions.* YZ and HZ designed the study, HZ, DH and JZ performed the experiments.
YZ and HZ analyzed the data, conducted model simulations, and wrote the paper. All other authors
contributed to discussion and writing.


*Competing interests.* The authors declare no conflict of interest.


*Acknowledgments.* Yue Zhao acknowledges the Program for Professor of Special
Appointment (Eastern Scholar) at Shanghai Institutions of Higher Learning.


*Financial support.* This work was supported by the National Natural Science Foundation
of China (grants 22022607, 21806104, and 42005090) and the Program for Professor of
Special Appointment (Eastern Scholar) at Shanghai Institutions of Higher Learning.

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
