# Peer review of "Direct probing of acylperoxy radicals during ozonolysis of α-pinene: constraints on"

_EGUsphere, 2023_

## Author Response (AR1)

**Contents in this file**

1. Responses to the referees' comments

2. Author's changes in the manuscript

3. A marked-up manuscript version showing the changes

We are grateful to the reviewer for the thoughtful comments on the manuscript. Our point-to-point responses to each comment are as follows (reviewer's comments are in black font and our responses are in blue font).

General Comments

In this manuscript the authors report a combined experimental and modelling study of the formation and fate of acylperoxy radicals formed from the reaction of a-pinene with ozone in a flow reactor. The alkyl and acyl $RO_2$ radicals and highly oxidized molecules (HOMs) were monitored using a chemical ionization mass spectrometer with nitrate ion ionization. $RO_2$ radicals and HOMS were assigned based on elemental formulas and acyl $RO_2$ radicals were distinguished from alkyl $RO_2$ radicals by addition of $NO_2$, which forms $RC(O)OONO_2$ (acyl peroxy nitrates) that are relatively stable under the conditions of the experiments, thus removing acyl $RO_2$ signal. Because the changes in acyl $RO_2$ concentrations can also impact other aspects of the chemistry, a detailed F0AM model employing a modified Master Chemical Mechanism was employed to interpret the results.

Overall, the experiments and modelling were well done and the approach seems to have yielded quite useful and interesting results. The authors provide a very thorough and thoughtful discussion of the results, which is clearly written and easy to follow. Considering the high technical quality of the study and the importance of these reactions to the formation of HOMs and ROOR dimers, both of which are currently of much interest because of their potential role in secondary organic aerosol (SOA) formation, I think the paper is well suited for ACP. I have only a few minor comments.

Specific Comments

1. Line 137: I don't understand the point of converting signals to "concentrations" using sulfuric acid since the actual concentrations will be highly sensitive to the structure of the $RO_2$ radical and HOM. Presenting the results this way is misleading. Since the "concentrations" are only used to calculate contributions of various species relative to each other, normalized signals will give the same results and be a more honest presentation of the data.

Response: Thanks for the reviewer's comment. We have changed the normalized concentrations of various species to their normalized signals in the revised manuscript.

2. Line 149: Have the authors considered partitioning of $RO_2$ radicals to particles and what influence that could have on the results? The vapor pressures of the radicals should be similar to those of HOMs, so I don't see any reason that they would not form SOA, and they likely undergo different reactions in the particles since isomerization would be restricted.

Response: We appreciate the reviewer's point. We used a scanning mobility particle sizer (SMPS, TSI) employing both long and nano differential mobility analyzers (model 3081 and 3085 for different particle sizes) to clarify whether there is SOA formation in the experiments. We did not observe SOA formation by SMPS in Exps 1-28. Only in Exp 31 where the reacted α-pinene reaches 36.8 ppb, we observed SOA formation with very low particle mass concentrations ($5.0\times10^{-4}$-$5.7\times10^{-3}$ μg m$^{-3}$) and number concentrations (63-395 # cm$^{-3}$). Therefore, we suggest that the negligible to low formation of SOA under these experimental conditions has no significant influence on the $RO_2$ fates.

We have added the results to Section 2.1 of the revised manuscript.

"To clarify whether there is SOA formation in the experiments, a scanning mobility particle sizer (SMPS, TSI), which consists of an electrostatic classifier (model 3080), a long or nano differential mobility analyzer (model 3081 and 3085 for different particle sizes), and a condensation particle counter (model 3087), was used to monitor the formation of SOA particles. Except in Exp 31 where the reacted α-pinene reached 36.8 ppb and there was low SOA formation with particle mass concentrations of $5.0\times10^{-4}$-$5.7\times10^{-3}$ μg m$^{-3}$ and number concentrations of 63-395 # cm$^{-3}$, no particle formation was observed by SMPS. Therefore, the $RO_2$ radicals and closed-shell products would be primarily distributed in the gas phase, with their fates negligibly influenced by the low SOA formation under these experimental conditions."

3. Line 342: In this section it is not clear to me what conclusions are based on measurements, modelling, or a combination of the two. Please make that more clear.

Response: To be more precise, we have clarified the relevant descriptions using "measured signals" and

"simulated contributions" in this section.

4. Line 530: Considering that $RO_2 + NO_2$ rate constants have been measured for a variety of alkyl and acyl $RO_2$ radicals and are pretty consistently ~1E–11 (Orlando & Tyndall 2012), it seems unlikely that the value is as low as suggested here. Any explanation based on $RO_2$ structure would imply that the same effects apply to the $RO_2 + NO$ rate constant, which is essentially identical to the $NO_2$ value (Orlando & Tyndall 2012). This would have significant consequences for predictions of conditions under which autoxidation reactions are important in the atmosphere, since this usually depends on the competition between $RO_2$ isomerization and the $RO_2 + NO$ reaction. What are other possible explanations for the apparent discrepancy?

Response: We appreciate the reviewer's point. Orlando and Tyndall (2012) have summarized that the rate coefficients of functionalized $RO_2 + NO_2$ are in the range of $(5\text{-}10)\times10^{-12}$ cm$^3$ molecule$^{-1}$ s$^{-1}$. However, these coefficients are mainly for the $RO_2$ species with small molecular sizes. A recent study by Berndt et al. (2015) determined a $RO_2 + NO_2$ rate coefficient of $(1.6\pm0.5)\times10^{-12}$ cm$^3$ molecule$^{-1}$ s$^{-1}$ for a highly oxidized acyl $RO_2$ radical $O,O-C_6H_7(OOH)_2O_2$ arising from the gas-phase ozonolysis of cycloalkanes, which is several times smaller the rates reported for the relatively simple $RO_2$ (Orlando and Tyndall, 2012). Therefore, it is possible that some of the α-pinene-derived $RO_2$ radicals react with $NO_2$ less efficiently than the smaller $RO_2$ radicals do. Such differences in the $RO_2 + NO_2$ rate coefficient may partially explain the observed increase in $C_{20}H_{34}O_x$ dimer formation as a function of added $NO_2$.

References:

Berndt, T., Richters, S., Kaethner, R., Voigtländer, J., Stratmann, F., Sipilä, M., Kulmala, M., and Herrmann, H.: Gas-Phase Ozonolysis of Cycloalkenes: Formation of Highly Oxidized $RO_2$ Radicals and Their Reactions with NO, $NO_2$, $SO_2$, and Other $RO_2$ Radicals, J. Phys. Chem. A, 119, 10336-10348, 10.1021/acs.jpca.5b07295, 2015.

Orlando, J. J. and Tyndall, G. S.: Laboratory studies of organic peroxy radical chemistry: an overview with emphasis on recent issues of atmospheric significance, Chem. Soc. Rev., 41, 6294-6317, https://doi.org/10.1039/C2CS35166H, 2012.

Response to Reviewer #2

We are grateful to the reviewer for the thoughtful comments on the manuscript. Our point-to-point responses to each comment are as follows (reviewer's comments are in black font and our responses are in blue font).

General Comments

In this study, the authors investigated the fraction of acyl peroxy radicals ($RO_2$) formed from alpha-pinene ozonolysis. Acyl-$RO_2$s are of crucial atmospheric importance due to their higher reactivity and their role in the formation of aerosol precursors. In flow reactor alpha-pinene ozonolysis experiments, $NO_2$ was used as an acyl-$RO_2$ scavenger, and the reduction in $RO_2$ signals was used to probe the fraction of acyl-$RO_2$s produced. The paper is well written and makes a significant contribution towards the better understanding of a key aerosol forming system in the atmosphere. I recommend the publication of the manuscript in ACP after the authors address my minor comments below:

In addition to dimer formation and producing RO, alkyl-$RO_2$ cross reactions also lead to ROH + R=O products, and if the inital peroxy radical group is on a primary carbon atom, the R=O can be a source of acyl-$RO_2$ following a secondary OH reaction. Alkyl $RO_2$s can have significant yields for this reaction. Was this accounted for in the model and in the analysis of the experiments?

Response: Thanks for the reviewer's comment. During the OH oxidation of α-pinene, acyl $RO_2$ can be formed from the secondary OH oxidation of aldehydes such as pinonaldehyde. However, in the present study, the secondary OH oxidation is significantly inhibited due to an excess of α-pinene compared to $O_3$, therefore the contribution of secondary OH oxidation to acyl $RO_2$ formation is expected to be relatively small. We considered the secondary OH oxidation in the model simulations, and found that the contribution of secondary OH oxidation to acyl $RO_2$ formation is negligible even under the high $O_3$ condition (Exp 22, 500 ppb α-pinene + 180 ppb $O_3$). As shown in Figure S9, the acyl $RO_2$ $C_9H_{13}O_4$ (C89CO3 in Figure S9a) and $C_{10}H_{15}O_5$ (C920CO3 in Figure S9b) can be formed from both C-C cleavage/H-shift of $C_{10}H_{15}O_4$-RO (Figure 4) and OH oxidation of the first-generation aldehyde products. However, the contributions from secondary OH oxidation are negligible for the two acyl $RO_2$ species during the whole reaction period under the high $O_3$ condition (Exp 22, 500 ppb α-pinene + 180 ppb $O_3$). In addition, the acyl $RO_2$ $C_{10}H_{15}O_4$ (C96CO3) that can be only formed from OH oxidation of pinonaldehyde contributes to only 0.01% and 0.2% of the total $C_{10}H_{15}O_4$-$RO_2$ and total acyl $RO_2$ concentration, respectively. Therefore, the contribution of secondary OH oxidation to acyl $RO_2$ in this study is minor and the majority of acyl $RO_2$ species measured here are formed from the ozonolysis channel.

[Figure]

Figure S9 Simulated contribution of different processes to the formation of (a) $C_9H_{13}O_4$ (C89CO3) and (b) $C_{10}H_{15}O_5$ (C920CO3) acyl-$RO_2$ during ozonolysis of α-pinene (Exp 22, 500 ppb α-pinene + 180 ppb $O_3$).

We have added the following discussions to Section 3.2 of the main text.

"In addition, the secondary OH oxidation of aldehyde products can also produce acyl $RO_2$ radicals during ozonolysis of α-pinene. However, in the present study, the secondary OH oxidation is expected to be insignificant due to an excess of α-pinene compared to $O_3$. Indeed, kinetic model simulations incorporating the secondary OH chemistry show that the contribution of secondary OH oxidation to acyl $RO_2$ formation is negligible even under high $O_3$ conditions (see details in Section S2 and Figure S9)"

We have also added the following content and Figure S10 to the SI.

"S2. Contribution of secondary OH oxidation to acyl $RO_2$ formation.

Considering that the secondary OH oxidation of aldehyde products can also contribute to the formation of acyl $RO_2$ during ozonolysis of α-pinene, kinetic model simulations incorporating secondary OH chemistry were also performed under typical experimental conditions. As shown in Figure S9, the acyl $RO_2$ $C_9H_{13}O_4$ (C89CO3 in Figure S9a,) and $C_{10}H_{15}O_5$ (C920CO3 in Figure S9b) can be formed from both C-C cleavage/H-shift of $C_{10}H_{15}O_4$-RO (Figure 4) and OH oxidation of the first-generation aldehyde products. However, the contributions from secondary OH oxidation are negligible for the two acyl $RO_2$ species during the whole reaction period. In addition, the acyl $RO_2$ $C_{10}H_{15}O_4$ (C96CO3) that can be only formed from OH oxidation of pinonaldehyde contributes to only 0.01% and 0.2% of the total $C_{10}H_{15}O_4$-$RO_2$ and total acyl $RO_2$ concentration, respectively (not shown). Therefore, the contribution of secondary OH oxidation to acyl $RO_2$ in this study is minor and the majority of acyl $RO_2$ species measured here are formed from the ozonolysis channel."

The more functionalized acyl-$RO_2$s with an -OOH group elsewhere in the molecule are known to undergo H-scrambling reactions to form peroxy acids (R-C(O)OOH) at rates of 1E3 – 1E5 s-1 (Knap et al. 2017, J. Phys. Chem. A, 121(7), pp.1470-1479). Can the authors comment on the possible role of this reaction in their experiments? For example, the ring-opened acyl $C_{10}H_{15}O_8$-$RO_2$ that they report has a 1,6 H-scramble available that leads to a peroxy acid and an alkyl $RO_2$. For their model system, Knap et al. estimate a rate coefficient for the 1,6 H-shift of 1.5E5 s-1. If the rate coefficient is comparable for the alpha-pinene derived $C_{10}H_{15}O_8$ acyl-$RO_2$ above, this could to an extent explain the low reduction in signal upon $NO_2$ addition.

Response: We appreciate the reviewer's point. We have performed a model simulation to evaluate the influence of H-scrambling reactions on the response of ring-opened acyl $C_{10}H_{15}O_8$-$RO_2$ to $NO_2$ addition. As shown in Figure S14, when a 1,6 H-shift rate of $1×10^5$ $s^{-1}$ is considered, the extent of the reduction in $C_{10}H_{15}O_8$-$RO_2$ with $NO_2$ addition indeed becomes smaller, especially when the yield of ring-opened $C_{10}H_{15}O_4$-$RO_2$ in the model is at a higher limit (89%). Therefore, the H-scrambling reactions of the ring-opened acyl $C_{10}H_{15}O_8$-$RO_2$ could to certain extent explain the low reduction in its signal upon $NO_2$ addition.

[Figure]

Figure S14 Simulated influence of H-scrambling reaction on the behavior of the ring-opened acyl $C_{10}H_{15}O_8$-$RO_2$ as a function of added $NO_2$ concentration (Exps 8-14). A 1,6 H-scrambling rate of $1×10^5$ $s^{-1}$ and an alkyl $RO_2$+$NO_2$ rate coefficient of $5×10^{-12}$ $cm^3$ $molecule^{-1}$ $s^{-1}$ were used in the model.

We have added the following discussions to Section 3.3 in the main text.

"Thirdly, the ring-opened $C_{10}H_{15}O_8$-$RO_2$, a highly functionalized acyl $RO_2$ radical with an –OOH group, may be able to undergo very fast intramolecular H-scrambling reactions to form a peroxy acid as suggested by Knap and Jørgensen (2017), which would compete with the $NO_2$ reaction and result in a lower reduction in its signal upon $NO_2$ addition (see details in Section S3)"

We have also added the following content and Figure S14 to the SI.

"S3 Possible influence of H-scrambling reactions on the behavior of $C_{10}H_{15}O_8$ acyl-$RO_2$

It has been suggested that the functionalized acyl $RO_2$ radicals with an -OOH group could undergo H-scrambling reactions to form peroxy acids at rates of $1\times10^3$-$1\times10^5$ s$^{-1}$ (Knap and Jørgensen, 2017). Here, we performed a model simulation to evaluate the influence of this reaction on the response of the ring-opened acyl $C_{10}H_{15}O_8$-$RO_2$ to $NO_2$ addition. As shown in Figure S14, considering a 1,6 H-shift rate of $1\times10^5$ s$^{-1}$, the simulated reduction in total $C_{10}H_{15}O_8$-$RO_2$ concentration with the addition of 30 ppb $NO_2$ decreases from 25% to 21% for a $C_{10}H_{15}O_4$-$RO_2$ yield of 30% (lower limit) and from 31% to 17% for a yield of 89% (higher limit). These results suggest that the H-scrambling reactions of the ring-opened acyl $C_{10}H_{15}O_8$-$RO_2$ could to certain extent explain the low reduction in its signal upon $NO_2$ addition."

Line 468: Regarding the speculation of the formation of alkyl $C_9H_{15}O_3$-$RO_2$ to explain the discrepancy between experiments and simulations, this would compete with the formation of the ring-opened and ring-retaining $C_{10}H_{15}O_4$-$RO_2$. How did the sensitivity analysis of including the $C_9H_{15}O_3$-$RO_2$ in the model affect the yield of the $C_{10}H_{15}O_4$-$RO_2$s and the subsequent acyl-$RO_2$s derived from them?

Response: We have evaluated the influence of the sensitivity analysis of $C_9H_{15}O_3$-$RO_2$ yield on other $C_{10}H_{15}O_4$-$RO_2$ as well as on the contribution of acyl $RO_2$ to total $C_{7-10}$ HOMs. As shown in Figure S16, as the $C_9H_{15}O_3$-$RO_2$ yield increases from 0% to 3%, the simulated concentration of a ring-retaining $C_{10}H_{15}O_4$-$RO_2$ radical (C10H15O4KB) decreases by only ~5% and other $C_{10}H_{15}O_4$-$RO_2$ species are basically unchanged. As the $C_9H_{15}O_3$-$RO_2$ is considered to only produce highly oxygenated alkyl $RO_2$ in the model, the increase in its yield results in a decrease in the contribution of acyl $RO_2$ to the total $C_9$ HOMs. However, the contributions of acyl $RO_2$ to total $C_7$, $C_8$, and $C_{10}$ HOMs are almost unchanged. These results indicate that the relatively small production of $C_9H_{15}O_3$-$RO_2$ has no significant influence on the yield of $C_{10}H_{15}O_4$-$RO_2$ and the subsequent acyl $RO_2$.

We have added the above results and discussion to Section 3.3 in the main text and Figure S16 to the SI.

[Figure]

Figure S16 Influences of $C_9H_{15}O_3$-$RO_2$ production on (a) the yield of $C_{10}H_{15}O_4$-$RO_2$ and (b) the contribution of acyl RO$_2$ to total C$_{7-10}$ HOMs (Taking Exp 8 as an example). The C10H15O4KB and C10H15O4RB denote a ring-retaining and a ring-opened C$_{10}$H$_{15}$O$_4$-RO$_2$, respectively (see Table S3 and the main text).

Are any of the acyl peroxy nitrates detected by the NO$_3$-CIMS? Alpha-pinene derived APNs with 8 oxygen atoms or more should have at least 2 -OOH functional groups and will presumably cluster well with NO$_3$. Do the decrease in e.g. C$_7$ and C$_8$ RO$_2$ signals when NO$_2$ is added show an increase in the corresponding APN signals? I think a spectrum figure maybe in the supplementary showing the acyl-RO$_2$ and acyl-ROONO$_2$ peaks would be useful.

Response: We have added a spectrum figure in SI showing the signal changes of acyl RO$_2$ and their corresponding RC(O)OONO$_2$ (Figure S3). It can be seen that the signals of acyl RO$_2$ decrease remarkably with the addition of NO$_2$. Accordingly, the signals of highly oxygenated RC(O)OONO$_2$ such as C$_9$H$_{13}$O$_9$NO$_2$, C$_9$H$_{17}$O$_7$NO$_2$, and C$_{10}$H$_{15}$O$_7$NO$_2$ increase significantly. We note that some of RC(O)OONO$_2$ have very similar m/z values with some alkyl RO$_2$. For example, the m/z values of C$_8$H$_{13}$O$_6$NO$_2$ (251.0641) and C$_8$H$_{13}$O$_8$NO$_2$ (283.0539) are very close to those of C$_9$H$_{15}$O$_8$-RO$_2$ (251.0767) and C$_9$H$_{15}$O$_{10}$-RO$_2$ (283.0665), respectively, which always have high ion signals. As a result, although some RC(O)OONO$_2$ are expected to be formed with NO$_2$ addition, they could not be unambiguously detected by nitrate-CIMS due to their overlapping with strong alkyl RO$_2$ peaks in this study.

We have added the following discussions to Section 3.1 in the main text.

"Along with the marked reduction in acyl RO$_2$ signals, the production of highly oxygenated RC(O)OONO$_2$ species such as C$_9$H$_{13}$O$_9$NO$_2$, C$_9$H$_{17}$O$_7$NO$_2$, and C$_{10}$H$_{15}$O$_7$NO$_2$ with the addition of NO$_2$ were observed (see the spectra in Figure S3). However, we note that although some RC(O)OONO$_2$ such as C$_8$H$_{13}$O$_6$NO$_2$ and C$_8$H$_{13}$O$_8$NO$_2$ are expected to be formed with NO$_2$ addition, they could not be unambiguously detected by nitrate-CIMS due to the overlapping of their peaks with strong alkyl RO$_2$ peaks (C$_9$H$_{15}$O$_8$-RO$_2$ and C$_9$H$_{15}$O$_{10}$-RO$_2$) in this study."

[Figure]

Figure S3 Signals of measured acyl RO$_2$ and the related RC(O)OONO$_2$ with and without the addition of NO$_2$ (Exps 8 and 14).

Figure 4. In pathway 3, the final H-shift of the acyl-oxy is unlikely to compete with CO$_2$ See reaction r12 and description therein in Vereecken et al. 2009, Phys. Chem. Chem. Phys. 11(40), pp.9062-9074.

Response: Thanks for the reviewer's comment. We have updated pathway 3 with a C$_{10}$H$_{15}$O$_3$ alkoxy radical that can undergo H-shift to form an acyl RO$_2$.

**Pathway 3**

Primary C$_{10}$H$_{15}$O$_4$  +RO$_2$  →  H shift  →  C$_{10}$H$_{15}$O$_5$

*Correspondence: Yue Zhao (yuezhao20@sjtu.edu.cn)

**Abstract**

Acylperoxy radicals ($RO_2$) are key intermediates in atmospheric oxidation of organic compounds and different from the general alkyl $RO_2$ radicals in reactivity. However, direct probing of the molecular identities and chemistry of acyl $RO_2$ remains quite limited. Here, we report a combined experimental and kinetic modelling study of the composition and formation mechanisms of acyl

$RO_2$, as well as their contributions to the formation of highly oxygenated organic molecules (HOMs)

during ozonolysis of α-pinene. We find that acyl $RO_2$ radicals account for 67%, 94%, and 32% of the highly oxygenated $C_7$, $C_8$, and $C_9$ $RO_2$, respectively, but only a few percent of $C_{10}$ $RO_2$. The formation pathway of acyl $RO_2$ species depends on their oxygenation level. The highly oxygenated acyl $RO_2$ (oxygen atom number $\geq 6$) are mainly formed by the intramolecular aldehydic H-shift (i.e., autoxidation) of $RO_2$, while the less oxygenated acyl $RO_2$ (oxygen atom number $< 6$) are basically derived from the C-C bond cleavage of alkoxy (RO) radicals containing an α-ketone group or the intramolecular H-shift of RO containing an aldehyde group. The acyl $RO_2$-involved reactions explain 50-90% of $C_7$ and $C_8$ closed-shell HOMs and 14% of $C_{10}$ HOMs, respectively. For $C_9$ HOMs, this contribution can be up to 30%-60%. In addition, acyl $RO_2$ contribute to 50%-95% of $C_{14}$-$C_{18}$

HOM dimer formation. Because of the generally fast reaction kinetics of acyl $RO_2$, the acyl $RO_2$ +

alkyl $RO_2$ reactions seem to outcompete the alkyl $RO_2$ + alkyl $RO_2$ pathways, thereby affecting the fate of alkyl $RO_2$ and HOM formation. Our study sheds lights on the detailed formation pathways of the monoterpene-derived acyl $RO_2$ and their contributions to HOM formation, which will help to understand the oxidation chemistry of monoterpenes and sources of low-volatility organic compounds capable of driving particle formation and growth in the atmosphere.

**1. Introduction**

Monoterpenes ($C_{10}H_{16}$) comprise an important fraction of nonmethane hydrocarbons in the global atmosphere (Guenther et al. 2012, Sindelarova et al. 2014) and make a significant contribution to the secondary organic aerosol (SOA) budget (Pye et al. 2010, Iyer et al. 2021). The presence of double bond and large molecular size of monoterpenes favor their oxidation reactivity towards $O_3$, hydroxyl (OH), and nitrate ($NO_3$) radicals (Berndt 2022, Roger et al. 2004, Atkinson, Hasegawa and Aschmann 1990, Kurten et al. 2015, Kristensen et al. 2016, Bianchi et al. 2019), as well as the formation of low-volatility products and SOA (Molteni et al. 2019, Shen et al. 2022, Bianchi et al. 2019, Zhang et al. 2018, Fry et al. 2014, Fry et al. 2009). The organic peroxy radicals ($RO_2$) in the gas-phase oxidation of monoterpenes can undergo autoxidation and form a class of highly oxygenated organic compounds (HOM) (Bianchi et al. 2019, Jokinen et al. 2014, Mentel et al. 2015, Berndt et al. 2016, Berndt 2022, Zhao, Thornton and Pye 2018, Bell et al. 2021), which are primarily low- or extremely low-volatility organic compounds (LVOCs and ELVOCs) (Bianchi et al. 2019, Ehn et al. 2014) and thus play a crucial role in SOA formation and growth.

Significant advances have been made in recent years concerning the monoterpene $RO_2$ autoxidation and its contribution to HOM formation (Zhao et al. 2018, Shen et al. 2022, Ehn et al. 2014, Berndt et al. 2016, Xu et al. 2019, Berndt 2022, Lin 
[revised manuscript text omitted]